# Intestinal Ketogenesis and Permeability

**DOI:** 10.3390/ijms25126555

**Published:** 2024-06-14

**Authors:** Anna Casselbrant, Erik Elias, Peter Hallersund, Erik Elebring, Jakob Cervin, Lars Fändriks, Ville Wallenius

**Affiliations:** 1Department of Surgery, Institute of Clinical Sciences, Sahlgrenska Academy at the University of Gothenburg, 40530 Gothenburg, Sweden; erik.elias@gu.se (E.E.); jajo0728@gmail.com (P.H.); lars.fandriks@gastro.gu.se (L.F.); ville.wallenius@gastro.gu.se (V.W.); 2Department of Microbiology and Immunology, Institute of Biomedicine, Sahlgrenska Academy at the University of Gothenburg, 40530 Gothenburg, Sweden; jakob.cervin@enera.com; 3Department of Surgery, Region Västra Götaland, Sahlgrenska University Hospital, 41345 Gothenburg, Sweden

**Keywords:** endotoxin method, high-fat diet, ketone bodies, permeability, small intestinal

## Abstract

Consumption of a high-fat diet (HFD) has been suggested as a contributing factor behind increased intestinal permeability in obesity, leading to increased plasma levels of microbial endotoxins and, thereby, increased systemic inflammation. We and others have shown that HFD can induce jejunal expression of the ketogenic rate-limiting enzyme mitochondrial 3-hydroxy-3-methylglutaryl-CoA synthase (HMGCS). HMGCS is activated via the free fatty acid binding nuclear receptor PPAR-α, and it is a key enzyme in ketone body synthesis that was earlier believed to be expressed exclusively in the liver. The function of intestinal ketogenesis is unknown but has been described in suckling rats and mice pups, possibly in order to allow large molecules, such as immunoglobulins, to pass over the intestinal barrier. Therefore, we hypothesized that ketone bodies could regulate intestinal barrier function, e.g., via regulation of tight junction proteins. The primary aim was to compare the effects of HFD that can induce intestinal ketogenesis to an equicaloric carbohydrate diet on inflammatory responses, nutrition sensing, and intestinal permeability in human jejunal mucosa. Fifteen healthy volunteers receiving a 2-week HFD diet compared to a high-carbohydrate diet were compared. Blood samples and mixed meal tests were performed at the end of each dietary period to examine inflammation markers and postprandial endotoxemia. Jejunal biopsies were assessed for protein expression using Western blotting, immunohistochemistry, and morphometric characteristics of tight junctions by electron microscopy. Functional analyses of permeability and ketogenesis were performed in Caco-2 cells, mice, and human enteroids. Ussing chambers were used to analyze permeability. CRP and ALP values were within normal ranges and postprandial endotoxemia levels were low and did not differ between the two diets. The PPARα receptor was ketone body-dependently reduced after HFD. None of the tight junction proteins studied, nor the basal electrical parameters, were different between the two diets. However, the ketone body inhibitor hymeglusin increased resistance in mucosal biopsies. In addition, the tight junction protein claudin-3 was increased by ketone inhibition in human enteroids. The ketone body β-Hydroxybutyrate (βHB) did not, however, change the mucosal transition of the large-size molecular FD4-probe or LPS in Caco-2 and mouse experiments. We found that PPARα expression was inhibited by the ketone body βHB. As PPARα regulates HMGCS expression, the ketone bodies thus exert negative feedback signaling on their own production. Furthermore, ketone bodies were involved in the regulation of permeability on intestinal mucosal cells in vitro and ex vivo. We were not, however, able to reproduce these effects on intestinal permeability in vivo in humans when comparing two weeks of high-fat with high-carbohydrate diet in healthy volunteers. Further, neither the expression of inflammation markers nor the aggregate tight junction proteins were changed. Thus, it seems that not only HFD but also other factors are needed to permit increased intestinal permeability in vivo. This indicates that the healthy gut can adapt to extremes of macro-nutrients and increased levels of intestinally produced ketone bodies, at least during a shorter dietary challenge.

## 1. Introduction

Consumption of a high-fat diet (HFD) has been suggested as a contributing factor behind increased intestinal permeability, leading to increased plasma levels of microbial endotoxins from the intestinal lumen and, thereby, increased systemic inflammation [1,2,3]. Such a widespread low virulent inflammation is a key event in secondary diseases associated with obesity, for example, diabetes type II and cardiovascular diseases [4,5,6,7,8,9,10]. In recent years, a growing body of evidence has suggested that the proximal digestive tract, under basic physiological conditions, participates in the regulation of key aspects of whole-body metabolism [11,12]. Specifically, lipid metabolism and lipid sensing mechanisms seem to be linked to regulating different aspects of metabolism [13,14,15]. These effects require intact signaling from the peroxisome proliferator-activated receptor alpha (PPARα) that regulates the central pathways of fatty acid oxidation (FAO), such as ketogenesis and beta-oxidation [16]. In addition to being involved in systemic energy metabolism, PPARα also regulates immunological processes in the gut mucosa, which is usually accompanied by epithelial barrier dysfunction [17]. In a previous study from our laboratory, we showed that an HFD resulted in markedly raised jejunal expression of the ketogenesis rate-limiting enzyme mitochondrial 3-hydroxy-3-methylglutaryl-CoA synthase (HMGCS2), resulting in the intestinal mucosa producing ketone bodies β-Hydroxybutyrate (βHB), not under starvation but under consumption of a diet with high-fat content [18,19]. Intestinal ketone body production has previously been described in suckling rats and mice pups, suggested to be part of a mechanism allowing large macromolecules, such as immunoglobulins, to pass over the mucosal barrier [20,21]. If intestinal ketogenesis and increased intestinal permeability are present in the infant, as recently shown also in obese adults [18], this suggests an exciting novel evolutionary mechanism that could explain why adult *Homo sapiens* might react with increased intestinal permeability in response to ingestion of HFD. However, the concept of permeability needs to be clarified. The permeability of an epithelial lining is determined mainly by two factors: paracellular and transcellular permeability, where the former depends on the intercellular junctional complex, e.g., tight junctions (TJs), while the latter determines transport through the enterocyte.

In the present study, the overall aim was to investigate the effect of dietary macro-composition on the expression of PPARα and whether intestinal ketone bodies produced in the proximal digestive tract after HFD affect paracellular or transcellular permeability. For that, we investigated the effects of 2-week fat- or carbohydrate-dominated diets in a cross-over study in healthy normal-weight volunteers. At the end of each two-week treatment, the subjects underwent a mixed meal test (MMT) and an endoscopy to retrieve jejunal biopsies to compare the fat- versus carbohydrate-exposed mucosa for permeability. To further elucidate functional permeability-induced mechanisms of interest, Caco-2 cells, mice, and human enteroids were used.

## 2. Results

### 2.1. PPARα Expression Is Altered by Diet

To determine if the diets had differential effects on the intestinal lipid metabolism signaling, the protein expression levels of PPARα were analyzed in the human jejunal biopsies. The expression level of PPARα was significantly reduced after HFD compared to the high-carbohydrate diet (HCD) (*p* = 0.008, n = 15 paired samples, Figure 1A). Previously we have shown that the expression levels of HMGCS2 synthase in the same biopsies are significantly increased after HFD compared to HCD [19]. As HMGCS2 is a rate-limiting enzyme for ketone bodies, we investigated the functional connection between PPARα and ketone bodies. Caco-2 cells were treated with the ketone body βHB in different concentrations (5–50 mM). After treatment, PPARα expression was significantly dose-dependently decreased (*p* = 0.03, Figure 1B). This indicates, to our knowledge for the first time, that accumulation of ketone bodies has a feedback inhibitory effect on PPARα expression.

### 2.2. No Sign of Inflammation after HFD in Non-Obese Healthy Individuals

Systemic inflammatory mediators were studied in blood samples. A complete list of blood biochemistry profiles after the two study periods has recently been published in *Nutrition* 2021 [22], including the CRP and ALP values. All values were within normal ranges in all fifteen participants. The CRP values were 0.13 ± 0.09 mg/L at baseline, 0.07 ± 0.07 mg/L after HFD, and <0 ± 0.00 mg/L after HCD. The ALP values were 1.03 ± 0.07 µkat/L at baseline, 0.94 ± 0.07 µkat/L after HFD, and 0.95 ± 0.07 µkat/L after HCD, respectively, and did not differ between the two diets. To study whether different diets are sufficient to induce postprandial endotoxemia, participants were given a mixed meal on day 12. Then, we evaluated the postprandial evolution of endotoxemia during 2 h. All values were low, some below the detection limit, and there were no differences between the diets. Dietary influence on inflammation and immune responses were also studied in the captured biopsy material. The expression of NFκβ, CD8α T Lymphocytes, and ALP did not differ between the diets. However, the expression of CD8β was significantly increased after HCD (*p* = 0.03, n = 15 paired samples, Figure 1E). These results indicate that we cannot see any general inflammation either systemically or in the intestinal proximal mucosa per se after 2 weeks of HFD in healthy volunteers.

### 2.3. Fat Is Mainly Localized to the Crypt

Endoscopic images after the different diets showed large differences despite an overnight fast. After the HFD, the jejunal mucosa showed a whitish spotted appearance giving the impression of lipid accumulation in the mucosa (Figure 2B). This appearance was not evident after the HCD, where the mucosae were pinkish and looked normal in their macroscopic appearance (Figure 2A). The mucosae were stained for fat using Oil Red O to see the distribution after the two diets. There was significantly more intense fat staining after a diet based on fat (*p* = 0.02, Figure 2E,F). The staining was localized to a greater degree to the intestinal epithelium at the bases of the villi. There were few lipid droplets scattered in the apical cytoplasm of the villus enterocytes, more under the nucleus or between the cells. Some lipid particles were also seen in the underlying lamina propria (Figure 2C–F). This means that the mucosa, despite fasting before the endoscopy, displays elevated fat accumulation after the HFD in healthy non-obese controls.

### 2.4. TJs Expression and Functional Analysis of Paracellular Permeability

Expression of TJ: Intraindividual TJ protein expression analysis after the respective diets was performed to identify indications of changes in paracellular permeability after HCD vs. HFD. None of the TJs studied were changed significantly between the diets in the intraindividual analyses (n = 15 paired samples) (Figure 3A–I). Immunofluorescence analysis confirmed the anatomical location of TJs being expressed between enterocytes in the lateral and basolateral membranes. Representative immunostaining for claudin-3 and Ck8 is shown in Appendix A. Ultrastructural analysis of TJs showed that the length of TJs was, on average, 218 nm at the top of intestinal villi and 200 nm at the base of villi and did not differ between the diets. The TJ diameter at the intestinal villi tips was, on average, 109 nm and at the bases 95 nm and was significantly wider at the top region (*p* = 0.004, n = 6 paired samples) but again did not differ between the diets (Figure 4A–D).

Functional analysis: Functional studies in the Ussing chamber using square wave pulse analysis are directly related to changes in paracellular conductance. PD and Rep were measured in the Ussing chambers for 30 min in all 15 individuals. The Iep was calculated using Ohm’s law. The PD ranged from −1.65 to −4.8 mV, the Rep from 5.0 to 14.4 Ω×cm^2^, and the Iep from 209 to 550 µA/cm^2^. None of the basal electrical parameters differed between the two diets (Table 1). After baseline measurement, the ketone body inhibitor hymeglusin (HG) was added. The Rep decreased over time but was significantly higher after 60 min in the presence of HG in preparations after HFD (*p* = 0.016, paired samples, Figure 5A). This effect could not be seen in mucosal samples after HCD. The PD also decreased marginally while the Iep increased slightly, but there was no significant difference between the diets.

To verify whether a HFD affects intestinal paracellular permeability, mice receiving HFD or NC for three weeks were used. HFD mice showed significantly higher weight gain compared with NC mice (*p* < 0.001). Previous mouse studies have shown that a three-week period of HFD increases the expression of HMGCS2 as well as levels of the ketone body βHB in the portal blood [18]. Mouse small intestine showed large “fat vacuoles” inside the cells of the villi after three weeks on HFD while control mice fed normal chow did not (Figure 5B,C). Despite major morphological changes in the intestinal mucosa, no basal differences in Rep were detected in the Ussing chambers between the different diets (HFD; 14.4 ± 1.1 vs. NC; 15.8 ± 2.0 Ω×cm^2^). After the baseline measurement, the ketone body inhibitor hymeglusin (HG) was added. The Rep decreased with time but was significantly higher after 60 min in the presence of HG in preparations after HFD (*p* = 0.002, n = 7–9, Figure 5D).

Ketone bodies and TJs expression: Human jejunal enteroid cultures were allowed to spontaneously differentiate into monolayers on semi-permeable membranes (Figure 5D). After differentiation, the TEER was above 800 ohm/cm^2^, and the membranes were considered to be fully confluent. Human enteroids were stimulated with butyrate, hymeglusin (HG), and ketone body βHB for 48 h. The TEER decreased after treatment in all groups but there was no difference between groups. The expression of the TJ protein claudin-3 increased significantly in enteroids treated with butyrate and hymeglusin in combination (*p* = 0.008, n = 13–14, Figure 5E). Taken together, all data suggest that we have no altered paracellular permeability after 2 weeks of HFD compared to HCD in healthy non-obese individuals. On the other hand, we have an elevated lipid depot in the mucosa and increased ketogenesis, which, ex vivo, can affect TJs and, thus, paracellular permeability.

### 2.5. Ketone Bodies Do Not Affect Transcellular Permeability

Transcellular permeability in Caco-2 cells: To investigate the functional connection between small intestinal mucosal ketone body production and transcellular permeability, we conducted Caco-2 cell-based experiments. In these experiments, large molecules (LPS and FD4) that have previously been described to pass the cell transcellularly were used [23]. First, we establish that Caco-2 cells grown on a membrane have the ability to produce ketone bodies, as we previously demonstrated [19]. Apical addition of the short-chain fatty acid butyrate for 48 h increased the basolateral concentration of the ketone body βHB. In the presence of the HMGCS2 inhibitor hymeglusin (HG), the concentration of βHB was significantly lower (*p* < 0.0001, n = 48–68, Figure 6A). Basolateral measurements of LPS showed no difference after treatment with butyrate or butyrate in combination with HG, where the LPS levels were low or below the detection limit (Figure 6B). Then, when we treated the Caco-2 cells for 48 h with a lipid cocktail mimicking a more realistic HFD consisting of both long and short fatty acids, the basolateral LPS levels increased significantly relative to untreated control (0.045 ± 0.004 pg/mL, *p* = 0.0007) but were at the same level in the presence of HG (0.061 ± 0.009 pg/mL, n = 4–8, Figure 6B). The probe permeation of FD4 also increased significantly after lipid cocktail stimulation (*p* = 0.0002, n = 6–10, Figure 6C). TEER did not differ before and after treatment. The Caco-2 cells stimulated with a lipid cocktail contained large vacuoles (stained black by osmium tetroxide) that are likely to contain fat (Figure 6D).

Transcellular permeability in mice: To verify whether a high-fat diet affects intestinal permeability, mice receiving HFD or NC for three weeks were used (see above). The probe permeability of FD4 in the gavage was measured in mesenteric lymph in an attempt to mimic the pathway of fat absorption and chylomicron transport from the intestine. The FD4-probe permeation in the lymph was significantly higher after four hours in mice on the HFD (*p* = 0.05, n = 2–8, Figure 6E) but was at the same level as NC after 24 h. Since 4 h seemed interesting, the addition of the ketone body inhibitor HG in the gavage was studied in a separate setup. The FD4-probe permeation was at the same level independent of treatment with the HG in HFD mice (n = 3–4, Figure 6F). These data suggest that the uptake of large molecules such as LPS and FD4-probe is facilitated by chylomicron-forming HFD. The uptake probably takes place in the same way as for fat, as the FD4-probe is found in the lymph. However, this transcellular permeation of LPS or FD4-probe was not ketone body-dependent as it could not be blocked by hymeglusin.

## 3. Discussion

This study addresses whether extremes in dietary macronutrient composition can affect intestinal permeability with a focus on HFD. Previous studies from our group have already shown, in the same clinical setting, that expression of the ketogenesis rate-limiting enzyme HMGCS2 increased in the jejunum after two weeks of HFD in healthy, normal-weight individuals compared to HCD [19]. Intestinal ketogenesis has been described by others to affect the uptake of macronutrients in suckling animals [20,21]. A Western diet (with a high content of fat) has been suggested to increase intestinal permeability [24]. The purpose of the present study was, therefore, to evaluate whether a diet high in fat, with a followingly increased intestinal ketogenesis, displays evidence of increased permeability compared to an equicaloric diet high in carbohydrates. In this cross-over study, we saw no permeability differences regarding CRP and ALP values, and neither did postprandial endotoxemia differ between the diets. The jejunal expression of the lipid metabolism signaling molecule PPARα, which regulates the central pathways of fatty acid oxidation such as ketogenesis, was, however, decreased after HFD. No basal electrical parameters ex vivo were changed, and neither was the expression of tight junction proteins. This indicated that the healthy human gut has the ability to adapt to extreme macronutrients, at least for a shorter period of time, and did not show any changes in permeability per se.

However, mucosal treatment with the ketone body inhibitor hymeglusin increased the resistance (Rep) in biopsies ex vivo after HFD. In addition, the expression of one of the major TJ proteins for mucosal density, the protein claudin-3, was increased when ketone body production was inhibited by hymeglusin in human enteroid monolayer cell cultures. These data indicate that intestinal ketone bodies have the ability to affect paracellular transports to some degree. Intestinal uptake of large molecules that pass transcellularly, such as LPS and FD4-probe, was analyzed in Caco-2 cells and in mice and was not found to be ketone body- or TJ-dependent. Instead, the uptake was facilitated by chylomicrons or “lipid droplets” formed in the intestinal epithelial cells in response to the HFD. These large fat particles will likely end up being transported from the small intestine by the lymph flow.

Previous studies have seen increased permeability and increased uptake of LPS in response to HFD [25,26]. However, these studies have been mainly performed in animals or obese subjects whose intestinal mucosa may have been exposed to high-energy diets for a long time [5,10]. One problem with measuring permeability in humans is that it is difficult to do this during the meal itself. All mucosal biopsies from human subjects are taken after 8–10 h of fasting, which likely affects the results. However, both the endoscopic images and the Oil Red O-staining showed lasting differences between the diets. Despite this, we could neither see any fat droplets apically in the enterocytes nor any differences in the expression of TJ proteins. The only strong, lasting difference we found in the mucosa after fasting was the reduced expression of PPARα. This was a surprising finding, as we previously described increased HMGCS2 in response to HFD in the same setting [19]. This is interesting since short-term lipid consumption has been described to increase PPARα expression in mice [27]. In another rodent study, direct stimulation with an intraperitoneal PPARα agonist has been shown to reduce food intake [28]. However, as consumption of a lipid-dominated diet over an extended period of time (two weeks in our experimental study) induces HMGCS2 expression and suppresses PPARα expression, this leads to an interesting paradox that short-term lipid consumption may be beneficial to glucose homeostasis by increasing PPARα. However, a more long-term lipid consumption could instead promote the development of obesity and obesity-associated co-morbidities by inducing expression of HMGCS2, leading to intestinal ketonegenesis, which may contribute to increased intestinal permeability as well as other detrimental effects, such as suppression of GLP-1 production, as we have shown earlier [19]. The present data from the Caco-2 cells also indicate that the accumulation of ketone bodies has a direct feedback inhibitory effect on PPARα expression, which could explain a temporal switch from beneficial to detrimental effects of prolonged dietary high fat exposure. In addition, as PPARα coordinates inflammatory signaling pathways, which may be accompanied by epithelial barrier dysfunction, we studied the expression of NFκβ and T Lymphocytes CD8α and CD8β [17]. Unlike Monteiro-Sepulveda et al. [29], who saw a significant increase in CD8αβ in the obese jejunal epithelium, we found no difference in NFκβ or CD8α. The expression of CD8β was, however, increased after HCD. ALP, which, in addition to regulating fat absorption, is involved in bacterial passage and endotoxin-induced inflammation, did not change between the diets [30].

The Ussing chamber measurement using square wave pulse analysis is directly related to changes in paracellular conductance. Since the large proportion of resistance over a cell layer is constant (consisting of apical, lateral, and basolateral membranes), a change is only related to the resistance between the cells determined mainly by two factors: intercellular junctional complexes and the dimension (length and width) of the intercellular spaces [31]. The junctional complexes are located at the lateral surface and subdivided into TJs and adherence junctions. This applies at least as long as the cell layer is intact; hence, the tissue is not analyzed for longer times (110 min in our case). In contrast, TEER measurement over cultured Caco-2 cells or enteroids cannot distinguish the resistances but is a good measure of cell viability and that the cells have grown into a dense and non-leaking cell layer. As mentioned, we saw no basal resistance difference after the diets in our healthy subjects. This was also confirmed in the mouse experiments despite the clear presence of fat droplets in the mucosa. However, mucosae exposed to a high fat content were ketone body sensitive, as the Rep increased in the presence of the ketogenesis inhibitor hymeglusin. In contrast, mucosae adapted to a high-carbohydrate diet were not sensitive to ketone bodies. This shows that the mucosa, despite no expression changes on TJs, had adapted to be more sensitive to HFD by synthesis of ketone bodies and that this can affect the paracellular permeability itself.

The endotoxin measurements on healthy individuals associated with the mixed meal test (MMT) were low to undetectable. These measurements are difficult, and there are many potential sources of error. One reason for the lack of results could be that the two-week dietary period is too short to show enough accumulation of endotoxins in the blood. Another possibility is that the “resistance” against mucosal effects of, e.g., HFD could be an inherent difference between lean and obese individuals. Endotoxin measurements have also been discussed a lot in the literature, as it is easy to get false negative or positive results [32]. This led us to develop our own method of measuring endotoxin (see Appendix A) for our in-house experiments. Because LPS has an affinity for chylomicrons [23], we hypothesized that chylomicron formation promotes LPS absorption. In agreement with our hypothesis, we found that Caco-2 cells released more cell-associated LPS (and FD4) after 48 h incubation with oleic acid, a long-chain fatty acid that induces chylomicron formation, than with butyric acid, a short-chain fatty acid that does not induce chylomicron formation. However, this transcellular chylomicron mechanism was not ketone body-sensitive as hymeglusin had no effect. In addition, FD4 and LPS were studied in mouse lymph in order to confirm the transport route for lipids. Certainly, we saw an increased amount of FD4 in the lymph after 4 h with HFD in mice, but after 24 h, the level was the same regardless of diet. Ketone bodies did not seem to affect this transcellular transport. The influence of ex vivo lipid micelles has previously been studied in human jejunal samples from obese and non-obese patients in Ussing chambers [5]. This study showed that the permeability of the FD4-probe did not differ basally between obese and non-obese subjects. However, after 3 h of stimulation with lipid micelles, the permeability of large probes increased in subjects with obesity, which could be derived from reduced expression of the TJ protein tricellulin. Thus, it appears that the mucosa of subjects with obesity is programmed to absorb more fat than the mucosa of a non-obese person. It could be due to an altered lipid signaling mechanism (already occurring after 2 weeks of HFD) as described above [13,14,15], but also the fact that we have a functionally active HMGCS2 enzyme that produces ketone bodies, e.g., βHB, in the mucosa of subjects with obesity on HFD [18]. Whether the jejunal mucosa in healthy individuals adapts differently to fat after two weeks compared to the subjects with obesity, future experiments will have to show. To further investigate the effect of βHB on paracellular permeability function, experiments were performed on human jejunal enteroids. These experiments clearly showed that preventing ketone body formation increased the expression of claudin-3, which is one of the most important TJ proteins for the mucosal barrier function.

In conclusion, we did not find any changes to intestinal permeability in vivo after two weeks of a high-fat or high-carbohydrate diet in healthy human volunteers. Further, neither the expression of inflammation markers nor the aggregate tight junction proteins were changed. This indicates that the gut of healthy human subjects has the ability to adapt to extremes of macro-nutrients over shorter periods of time. However, ex vivo, we found evidence for regulation of the PPARα and tight junctions by the ketone body βHB, indicating that it has signaling properties related to permeability regulation in the intestinal mucosa. In the future, studies on mucosa from subjects with obesity are needed in order to evaluate whether there are inherent differences in the intestinal adaptation mechanisms to different macronutrient compositions in regard to intestinal permeability.

## 4. Material and Methods

### 4.1. Subjects and Ethics

Fifteen healthy volunteers (7 females) who were not taking any medications, mean age of 25.5 [23,24,31,33,34,35,36] years, BMI~23 kg/m^2^, were recruited to the study. The study procedures were performed in accordance with the Declaration of Helsinki and approved by the Regional Ethical Review Board of Gothenburg with approval number 807-11. The study was registered at ClinicalTrials.gov (NCT02088853) and was performed at the Dept. of Surgery, Sahlgrenska University Hospital, Gothenburg, between February and December 2014. The primary outcome measure in ClinicalTrials.gov was defined as mucosal surface enlargement factor, and the secondary outcome measures were defined as epithelial electrical current and mucosal electrical resistance in vitro, as well as glycemic control following a mixed meal test (MMT). All individuals were informed verbally and in writing and signed an informed consent form before being included in the study. Three reports have already been published regarding this cross-over study. First, a report on the systemic glucose clearance following an MMT [22], a second report on glucose transport in jejunal biopsies [19], and a third morphological report on mucosal adaptation in relation to the dietary intervention [33].

### 4.2. Diet Intervention

The diets given were composed so that 60% of the energy would come either from lipids or from carbohydrates, as previously described [22]. Each diet period lasted two weeks, and the starting order was randomized in blocks for logistic reasons. After the initial two weeks, the subjects underwent an endoscopy with the collection of mucosal biopsies. The subjects then had a “wash-out period” of at least two weeks, during which they were instructed to eat their habitual diet. After this, the diets were switched in a cross-over fashion for another two weeks, with renewed endoscopy and mucosal tissue sampling at the end. All food consumed by the participants was handed out from the study center.

### 4.3. Mixed Meal Test and Blood Samples

At the end of each two-week diet period (day 12), the participant visited the laboratory after an overnight fast to do an MMT. Fasting blood samples were obtained via an intravenous forearm cannula. The study participants were served a 600 kcal brunch (15 E% protein, 31 E% carbohydrates, 54 E% fat) followed by blood sampling at 15, 30, 45, 60, 90, and 120 min after the start of the meal. Blood samples were prepared for plasma and serum, aliquoted, and directly frozen in liquid nitrogen and stored at −80 °C until analysis. For endotoxemia measurements, blood samples were collected in glass tubes containing sodium heparin (BD Diabetes, Erembodegem, Belgium). C-reactive peptide (CRP) (sensitive), alkaline phosphatase (ALP), and endotoxemia biochemistry profiling were analyzed by the central laboratory of the Sahlgrenska University Hospital (accredited according to European norm en45001).

### 4.4. Limulus Amoebocyte Lysate Assay

Endotoxemia in plasma was determined by using the Limulus amoebocyte lysate (LAL) assay in kinetic chromogenic conditions (Pierce ^TM^ LAL Chromogenic Endotoxin Quantitation Kit, Cat.nr. 88282, ThermoFisher Scientific, Stockholm, Sweden). In this method, the time at which the optical density (OD) starts to increase significantly is negatively correlated with the sample endotoxemia. The analysis and all the steps were run according to the protocol with 10× diluted samples using pyrogen-free water. After incubation with LAL, the substrate solution, followed by a stop reagent (25% acetic acid), was added to each sample. The results were analyzed with a spectrophotometer with a wavelength of 405 nm. The limit of sensitivity of the kinetic chromogenic assay is 0.005 EU/mL; therefore, endotoxemia down to 0.05 EU/mL can be detected in plasma considering a 1:10 dilution.

### 4.5. Endoscopy

On day 14 of each dietary period, the participants returned to the hospital after an overnight fast. The study participants were given routine premedication (midazolam and an opiate intravenously) immediately before the endoscopy. Under ocular control, a pediatric 130 cm enteroscope was introduced through the mouth via the pyloric channel into the duodenum and further ~50 cm into the jejunum. Eight to ten biopsies were taken from the jejunal mucosa and were either snap-frozen or chemically fixated for later expression analysis, as described below. The remaining biopsies were prepared for functional assessments in mini-Ussing chambers.

### 4.6. Ussing Chamber Experiments

Mucosal biopsies were immediately immersed in ice-cold oxygenated Krebs solution containing (in mM) 118.07 NaCl, 4.69 KCl, 2.52 CaCl_2_, 1.16 MgSO_4_, 1.01 NaH_2_PO_4_, 25 NaHCO_3_, and 11.10 glucose. The mucosa was mounted in mini-Ussing chambers that had a biopsy insert with a diameter of 2 mm and an area of 0.034 cm^2^ (Warner Instruments, Hamden, CT, USA). Both the luminal and the serosal sides were bathed in 5 mL Krebs solution, continuously oxygenated, and stirred with 95% O_2_ and 5% CO_2_ gas flow at 37 °C. In general, 6 Ussing chambers were mounted per individual. Potential difference (PD) was measured with a pair of matched calomel electrodes (REF401, Radiometer Analytical, Denmark), and the epithelial electrical resistance (Rep) was assessed by use of the Ussing Pulse Method. The latter has the advantage of estimating specifically paracellular permeability and is described in detail elsewhere [31,34]. Epithelial current (Iep) is then calculated using Ohm’s Law, where Iep = PD/Rep. In the present setup, data sampling and pulse inductions were computer-controlled using specially constructed hardware and software developed in LabView (https://www.ni.com/en/shop/labview.html (accessed on 11 June 2024) National Instruments, Austin, TX, USA).

Experimental procedures: After an equilibration period of 20 min, basal parameters were measured over 30 min. Then, the ketone body inhibitor hymeglusin (HG) 1 µM was added, and the electrical date was measured for an additional 60 min.

### 4.7. Western Blot Analyses

Frozen human jejunal mucosa samples were sonicated in ice-cold protein extraction buffer (10 mM potassium phosphate buffer, pH 6.8, containing 1 mM ethylenediaminetetraacetic acid, 10 mM 3-[(3-cholamidopropyl) dimethylammonio]-1-propanesulfonate) and Complete™ protease inhibitor cocktail. After sonication, each homogenate was centrifuged (10,000× *g*, 10 min, 4 °C), and the supernatant protein content was quantified with the standard Bradford method. Total protein samples were diluted in SDS buffer and heated at 70 °C for 10 min before loaded on NuPage 10% Bis-Tris gels, and the electrophoresis was run using MOPS buffer. After electrophoresis, proteins were transferred to polyvinyldifluoride membranes using the iBlot blotting system. Membranes were incubated with primary antibody against claudin-1 (1:250, 51-9000, Invitrogen, Waltham, MA, USA), claudin-2 (1:200, 51-6100, Invitrogen), claudin-3 (1:500, 34-1700, Invitrogen), claudin-4 (1:250, 32-9400, Invitrogen), claudin-15 (1:500, HPA053199, Atlas Antibodies, Stockholm, Sweden), occludin (1:1000, 71-1500, Invitrogen), zona-occludin-1 (ZO-1) (1:300, 61-7300, Invitrogen), cytokeratin-8 (Ck8) (1:500, ab53708, Abcam, Cambridge, UK), tricellulin (1:250, 48-8400, Invitrogen), nuclear factor kappaβ (NFkβ) (1:500, AJ1537a, NordicBiosite, Stockholm, Sweden), alkaline phosphatase (ALP) (1:500, ab97532, Abcam), CD8α (1:100, MA5-14548, Invitrogen), CD8β (1:200, sc-25277, Santa Cruz Biotechnology, Santa Cruz, CA, USA), and PPARα (1:200, sc-9000, Santa Cruz Biotechnology) followed by a HRP-linked secondary antibody (1:2000, #7074 or #7076, Cell Signaling Technology, Leiden, The Netherlands). Chemiluminescense was measured after the addition of WesternBright Quantum reagents (K-12042, Advansta Corporation, Menlo Park, CA, USA). Images were captured by a Chemidoc™ XRS system, and semi-quantification was performed using Quantity One software (Version 6.0.1, BioRad Laboratories, Hercules, CA, USA). The membranes were sequentially stripped using ReBlot Plus Mild solution (Millipore, Temecula, CA, USA). Glyceraldehyde 3-phosphate dehydrogenase (GAPDH) (1:1000, IMG-5143A, Imgenex, San Diego, CA, USA) was used as control for equal loading for each tested sample. Data are presented as the ratio between the optical density of the primary antibody and the GAPDH in each sample.

### 4.8. Immunohistochemistry and Oil Red O

Biopsies were fixed in phosphate-buffered 4% formaldehyde, dehydrated, embedded in paraffin, and cut into 5 μm sections. For immunostaining, sections were rehydrated, and antigens were retrieved by boiling for 20 min in 10 mM citrate buffer (pH 6.0), blocked in 5% normal goat serum, and incubated in primary antibody overnight at 4 °C (see above). After primary antibody incubation, slides were washed and incubated with secondary antibody for 2 h in darkness at room temperature. After washing, slides were counter-stained with Hoechst staining and cover-slipped with ProLong Gold anti-fade reagent (P36930, Invitrogen). Blocking buffer instead of primary antibody was used as a negative control. The slides were analyzed using a fluorescence microscope (Nikon Eclipse E400, Tokyo, Japan). For visualization of fat incorporation, frozen human jejunal samples were processed following standard practice at the Clinic of Pathology at Norra Älvsborgs Länsjukhus in Sweden (accredited according to Swedish Norm), cut into 6–8 µm sections, and stained using Oil Red O (0.5% solution) according to the manufacturer´s (BDH chemicals) instructions. Slides were analyzed by PH in a coded fashion.

### 4.9. Transmission Electron Microscopy

Samples from 6 individuals were further processed for ultrastructural analyses. The biopsies were taken between 7 and 8 a.m. (before the morning meal) and kept in ice-cold Ringer solution for ~30 min before fixation in ice-cold Karnovsky fixative (2% formaldehyde, 2.5% glutaraldehyde, 0.02% Na-azide in 0.05 M sodium cacodylate, pH 7.2). After fixation overnight, the specimens were post-fixed in 1% OsO_4_ and 1% K_4_Fe(CN)_6_ in 0.05 M sodium cacodylate for 2 h at 4C. A bloc staining followed in 0.5% uranyl acetate in distilled water for 1 h at room temperature in darkness. The tissues were then rinsed in distilled water, dehydrated in rising concentrations of ethanol followed by 100% acetone, infiltrated, and embedded in Agar 100 resin. Then, 60 nm ultramicrotome sections were photographed in an electron microscope at primary magnifications ×4000 or ×8000, focusing on the apical parts of enterocytes, from the base and from the tops of the villi. The TJ length and width were measured in cells with a random start, and in each individual up to 6 cells were measured at each level and each diet. All sections were coded until after all measurements had been concluded.

### 4.10. Caco-2 and Human Enteroid Monolayer Cell Cultures

Caco-2 cells at passages 48–52 were seeded onto 12-well transwell membranes (3.0 μm pore size, Corning, Corning, NY, USA) and expanded in expansion media (high glucose concentration, 25 mM) Dulbecco’s modified Eagle medium (DMEM, Invitrogen), 10% fetal bovine serum (Invitrogen), 1X non-essential amino acids (NEAA, Invitrogen), 1X penicillin–streptomycin (PEST, Invitrogen) on both sides of membrane until confluency. When confluent, cells were differentiated into a small intestine-like monolayer by culturing in high glucose, serum-free (SF) media (high glucose DMEM, 1X NEAA, 1X PEST) on the apical side and insulin–transferrin–selenium (ITS) media (high glucose DMEM, 1X ITS (Invitrogen), 1X NEAA, 1X PEST) on the basolateral side of the membrane for 14 days. Transepithelial electrical resistance (TEER) was measured using a voltmeter (Merck Millipore, Darmstadt, Germany) during differentiation.

Human enteroid cultures were established from an endoscopic biopsy taken from the jejunal portion of the small intestine from a healthy volunteer (female, 23 years old, ethics application number 049-16) using the technique developed by Sato et al. 2011 [35]. In brief, isolated crypts were seeded and embedded in Matrigel (Corning) in 24-well plates and cultured with Wnt3A-supplemented expansion media (IntestiCult Organoid Growth Media (STEMCELLS, Vancouver, BC, Canada), Y-27632 10 μM (Sigma-Aldrich, St. Louis, MO, USA), gentamycin 0.1% (Invitrogen) to enrich stem cells. To expand, Matrigel encapsulated cultures were treated with ice-cold Cell Recovery Solution (Corning) on ice for 1 h before being centrifuged and washed. Cystic cultures were broken into smaller fragments by pipetting before being seeded into Matrigel again. Expanded cultures passage 8–12 were recovered as previously described and seeded onto collagen-IV coated (15 μg/cm^2^, Sigma-Aldrich) 24-well transwell membranes (0.4 μm pore size, Corning) and expanded in expansion media on both sides of the membrane until confluent. When confluent, media were changed to differentiation media (1:1 IntestiCult Organoid Growth Media component A: DMEM/F12 with GlutaMAX (Invitrogen) and 15 mM HEPES (Invitrogen), 0.1% gentamycin (Invitrogen)) for 7 days. During differentiation, TEER was measured using a voltmeter (Merck Millipore).

Experimental procedures Caco-2: Differentiated monolayers of Caco-2 cells were cultured for 48 h in low-concentration glucose (5.5 mM) media (SF and ITS) with different combinations of additives: short-chain fatty acid butyrate 10 mM (Sigma-Aldrich), HMGCS2 inhibitor hymeglusin 1μM (Santa Cruz Biotechnology, Dallas, TX, USA), β-Hydroxybutyrate 5–50 mM (Sigma-Aldrich), FD4 probe 2 mg/mL, and LPS 1 µg/mL (Sigma-Aldrich). In some experiments, the Caco-2 cells were stimulated with a cocktail of fats: cholesterol 0.05 mM, 2-monooleoylglycerol 0.2 mM, L-∝-lysophosphatidyl choline 0.2 mM, and oleic acid 0.6 mM, as previously described [36]. After 48 h, basolateral media were collected, and βHB concentrations were quantified according to a β-Hydroxybutyrate Colorimetric Assay Kit (Cayman Chemicals, Ann Arbor, MI, USA). The concentration of the FD4 probe was spectrophotometrically assessed at excitation 480 nm and emission 535 nm (TECAN, Salzburg, Austria), and concentrations of LPS were measured using an in-house method (see Appendix A). In some experiments, cells were saved for morphological analysis.

Experimental procedures enteroids: Differentiated monolayers from human enteroids were cultured for 48 h in low-concentration glucose (5.5 mM) media (DMEM) with different combinations of the short-chain fatty acid butyrate 10 mM, HMGCS2 inhibitor hymeglusin 10 μM, and β-Hydroxybutyrate 10 mM (Sigma-Aldrich). After 48 h, proteins were extracted from cells for Western blot analysis.

#### 4.10.1. Mice Experiments

The 4–5-week-old male and female C57BL/6J mice were purchased from Harland (Horst, The Netherlands). Animals were maintained under controlled temperature (23 °C) and light conditions (lights on, 6:30–18:30 h), with ad libitum access to water and pelleted food. Male and female mice were fed HFD or normal chow (NC) for 3 weeks. The HFD consisted of pelleted food (Surwit Diabetogenic Rodent Diet, D12309; Research Diets, Inc., New Brunswick, NJ, USA) and had the following nutrient content in g%: fat 35.9%, protein 23%, carbohydrate 35.5%; total calorific content: 5.6 Kcal/g.

After a 2 h fast, all mice received an oral gavage with 0.5 mL intralipids, including FD4-probe (2 mg/mL) and HMGCS2 inhibitor hymeglusin (10 μM). After 20 min, 4 or 24 h, the mice were anesthetized with isoflurane. The abdomen was opened, the mesenteric lymph duct was identified and canulated, and lymph was collected. Samples were kept on ice, and then the concentration of FD4-probe in the lymph was analyzed as described above. In some animals, mucosa from the small intestine was collected and used for permeability analyses in Ussing chambers as well as fixated for later morphological analysis.

#### 4.10.2. Data Analysis and Statistics

Friedman´s and Wilcoxon’s signed rank test for related variables and ordinary one-way ANOVA and Mann–Whitney test for independent variables were used where appropriate. Data are presented as mean ± SEM. A *p*-value of ≤0.05 was considered significant. All analyses were performed using Prism 9 for Mac OS (GraphPad, San Diego, CA, USA).

## Figures and Tables

**Figure 1 ijms-25-06555-f001:**
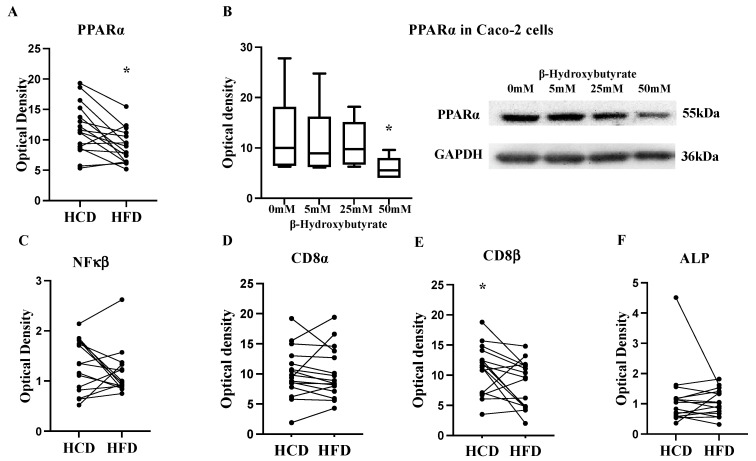
(**A**) The protein expression levels of peroxisome proliferator-activated receptor alpha (PPARα) was significantly decreased in the jejunum of healthy volunteers (n = 15) after a two-week high-fat diet (HFD) compared to a high-carbohydrate diet (HCD). (**B**) Caco-2 cells incubated with the ketone body β-hydroxybutyrat (5, 25, and 50 mM) during 48 h inhibited PPARα protein expression (n = 6 in each group). (**C**) Protein expression levels of NFκβ, (**D**) CD8α, (**E**) CD8β, and (**F**) alkaline phosphatase (ALP) from the human cross-over study. Data are related to a housekeeping protein, GAPDH. GAPDH, glyceraldehyde-3-phosphate dehydrogenase; OD, optimal density (Wilcoxon’s signed rank test and Mann–Whitney test). * *p* < 0.05.

**Figure 2 ijms-25-06555-f002:**
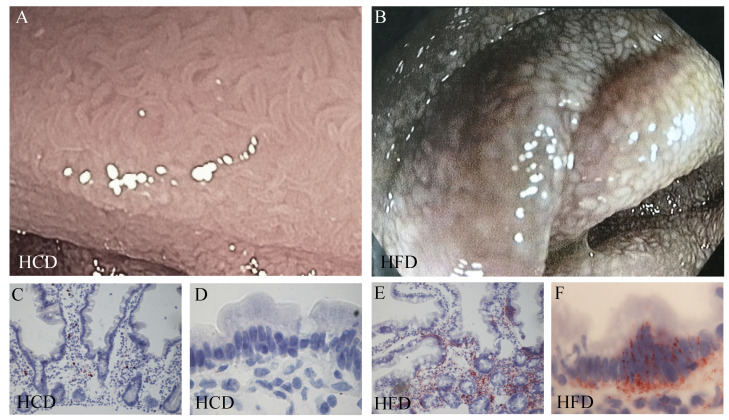
Representative endoscopic image of human jejunal mucosa after a two-week high-carbohydrate diet (HCD) showing normal pinkish appearance (**A**) or after a high-fat diet (HFD) with a whitish spotted appearance (**B**). Oil Red O staining showing fat distribution in jejunal mucosal biopsies obtained after two weeks of HCD (**C**,**D**) or HFD (**E**,**F**). The fat staining (red) is primarily located at the bases of the villi and is mainly found under the nucleus or basolaterally between the cells. Original magnification ×10 (**C**,**E**) and ×40 (**D**,**F**).

**Figure 3 ijms-25-06555-f003:**
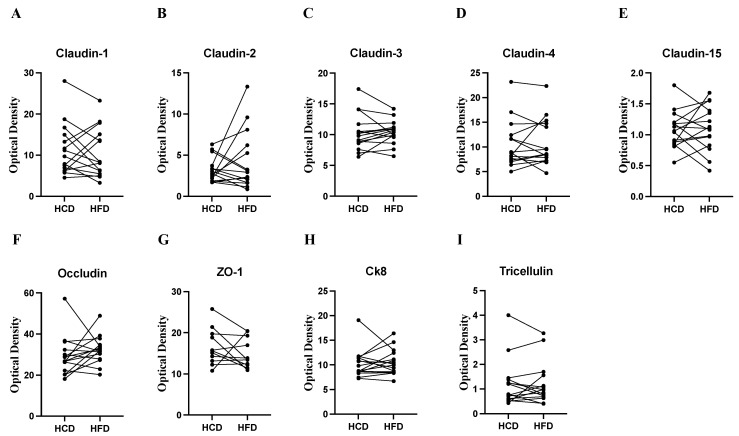
Western blot analysis of tight junction proteins claudin 1, 2, 3, 4, and 15, occludin, zona-occludin 1 (ZO-1), cytokeratin 8 (Ck8), and tricellulin in human jejunal mucosal biopsies obtained after 2 weeks of a high-carbohydrate diet (HCD) or high-fat diet (HFD) (cross-over design, n = 15). None of the analyzed protein expressions differed between the diets. Data are related to the housekeeping protein GAPDH (glyceraldehyde-3-phosphate dehydrogenase), and paired samples are reported (Wilcoxon’s signed rank test).

**Figure 4 ijms-25-06555-f004:**
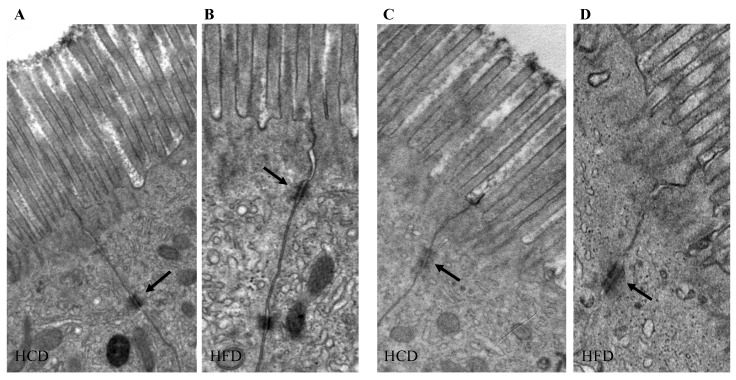
Ultrastructural analysis of human jejunal mucosa. The figure shows the length and diameter of tight junctions (arrows) between surface enterocytes at the top (**A**,**B**) and at the base (**C**,**D**) of intestinal villi after a two-week high-carbohydrate diet (HCD) or high-fat diet (HFD) (n = 6). The length of tight junctions was the same at the top or base of the villi and was not affected by the diets. The tight junction diameter was significantly wider at the top region but did not differ between the diets. Original magnification ×8000.

**Figure 5 ijms-25-06555-f005:**
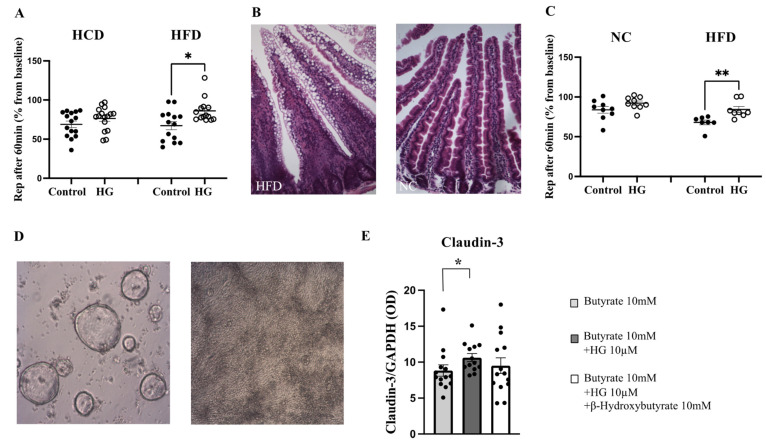
(**A**) Jejunal mucosa from the human cross-over study were mounted in the Ussing chamber and treated with the ketone body inhibitor hymeglusin (HG) for 60 minutes. HG significantly improved the epithelial resistance (Rep) in preparations after a high-fat diet (HFD) but not in mucosal preparations after the high-carbohydrate diet (HCD) (n = 15). (**B**) Mouse small intestine shows large fat droplets after 3 weeks of HFD or normal appearance after normal chow (NC). (**C**) Mouse small intestine mounted in the Ussing chamber treated with HG for 60 minutes significantly increased Rep in preparations after HFD but not in preparations after NC (n = 7–9). (**D**) Phase contrast micrograph of human jejunal enteroids ex vivo culture in cystic (left) and differentiated monolayer (right). (**E**) Enteroids stimulated with butyrate, the inhibitor HG, and β-Hydroxybutyrate for 48 h. The tight junction protein claudin-3 increased significantly in enteroids stimulated with butyrate and HG (n = 13–14). Data are given as means ± SEM (Mann–Whitney test). * *p* < 0.05, ** *p* < 0.01.

**Figure 6 ijms-25-06555-f006:**
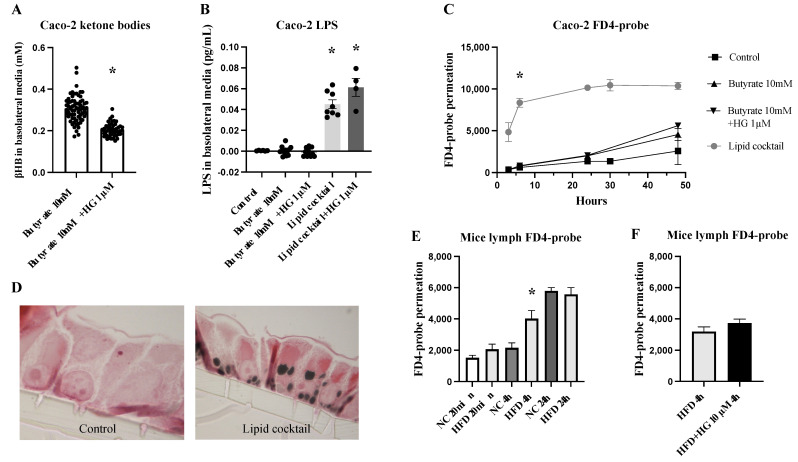
Quantification of β-hydroxybutyrat (βHB), LPS, and FD4-probe in basolateral media after 48-hour stimulation with butyrate, hymeglusin (HG), or a “cocktail of lipids” in Caco-2 cells. (**A**) HG significantly inhibited βHB-ketone body formation in Caco-2 cells (n = 48–68). (**B**) Treatment with butyrate or HG failed to increase LPS transport (n = 6–12). However, a lipid cocktail increased LPS transport, which could not be blocked by HG (n = 4–8). (**C**) FD4-probe permeation was significantly larger after stimulation with a lipid cocktail (n = 6–10). (**D**) Caco-2 cells that received the lipid cocktail showed large “fat droplets” inside of the cells. (**E**) FD4-probe permeation in lymph was significantly higher after 4 h in mice on HFD compared to normal chow (NC) (n = 2–8). (**F**) FD4-probe permeation into lymph was equally independent of HG treatment in HFD mice after 4 h (n = 3–4). Data are given as means ± SEM (Mann–Whitney test). * *p* < 0.05.

**Table 1 ijms-25-06555-t001:** Basal electrical parameters in Ussing chambers from a human cross-over study on jejunal mucosa after a 2-week HCD or HFD.

	PD (mV)	Rep (Ω×cm^2^)	Iep (µA/cm^2^)
HCD			
(n = 15, N = 60)	−3.15 ± 0.2	7.45 ± 0.2	437 ± 18.3
HFD			
(n = 15, N = 60)	−2.74 ± 0.2	8.34 ± 0.7	366 ± 28.3

Abbreviation: HCD; high-carbohydrate diet, HFD; high-fat diet, PD; potential difference, Rep; epithelial electrical resistance, Iep; epithelial electrical current, n; number of individuals, N; number of preparations. Values are given as mean ± SEM.

## Data Availability

All authors had access to the study data and reviewed and approved the final manuscript. All data relevant to the study are included in the article or uploaded as Appendix A. Individual participant data will not be shared.

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
