# Peer review of "Intestinal Ketogenesis and Permeability"

_ijms, 2024, doi:10.3390/ijms25126555_

Round 1

Reviewer 1 Report

Comments and Suggestions for Authors

Dear author/authors,

I am pleased to review the manuscript (MS) titled “Signaling Properties and Permeability Regulation of Mucosal Ketone Bodies in Human Intestinal Mucosa”. In general, the manuscript contains original and valuable research information. The manuscript is well written and organized, obtaining relevant results that can be applied to implement effective management of human gut physiology, especially high exposure of macro-nutrients and increased levels of intestinally produced ketone bodies. However, the number of samples and duration of the experiment are the major issues in this research. It is also needed to reduce the similarity rate. Some texts are copied and need to be reorganized the sentences. Also, some rigorous English and references check, and some contents are also necessary. The manuscript could be accepted after carefully addressing the issues and also looking into the manuscript.

Thank you

Comments on the Quality of English Language

Some minor English editing is required.

Author Response

AUTHORS RESPONSE TO REVIEWER #1

Reviewer 1.
1. The number of samples and duration of the experiment are the major issues in this research.

AUTHORS:

  1. We thank the Reviewer 1 for giving valuable criticism and advice. We have now clarified the number of samples and duration of the experiments in the text marked yellow at page 13 (lane 22), 15 (lane 7), 16 (lane 4 and 25), 17 (lane 6, 15, 16 and 21), 18 (lane 3, 4, 11-12, 22 and 24), 19 (lane 3, 6, 7-8, 17 and 19), 20 (lane 5), 22 (lane 17), 23 (lane 14), 31 (lane 3, 5 and 7), 33 (lane 4-5), and 35 (lane 3, 7 and 8).

AUTHORS RESPONSE TO REVIEWER #1

Reviewer 1.
1. Some texts are copied and need to be reorganized in the sentences. Also, some rigorous English and references check, and some contents are also necessary.

AUTHORS:

  1. We thank the Reviewer for giving valuable criticism and advice. We have now carefully gone through the manuscript and changed/clarified some sentences and removed duplications according to reviewer´s suggestion (see markings in the text). We have also clarified some results presented.

Reviewer 2 Report

Comments and Suggestions for Authors

In this manuscript, the authors investigate whether ketone byproducts of a high-fat diet (HFD) may alter the intestinal barrier. Evidence is provided that: i) the metabolic regulator PPAR-alpha is reduced by ketone bodies generated by HFD, ii) an inhibitor of ketogenesis increases the expression of tight junction proteins and the trans-epithelial intestinal barrier. With the chosen 2/3-week exposure to HFD, no significant changes in intestinal permeability were observed, both in the experimental and clinical settings. This study is based on a sound rationale and an accurate analysis of the relevant literature, the experiments have been carefully designed and performed, and the results are convincing and thoroughly discussed.

Minor points

1) For proper comparison, an endoscopic image of control HCD mucosa showing no white spots should be added to Fig. 2.

2) Figure 4, ultrastructural tight junction (TJ) analysis. At lines 359 and 372, please specify ‘intestinal’ villi. At first reading, focusing on the images shown in Figure 4, I understood ‘microvilli’ and did not figure out what the author meant in their description.

3) Line  397: unclear sentence.

4) Line 442: osmium tetroxide?

Author Response

AUTHORS RESPONSE TO REVIEWER #2

Reviewer 2.
1. For proper comparison, an endoscopic image of control HCD mucosa showing no white spots should be added to Fig. 2.

AUTHORS:

  1. We thank the Reviewer for giving valuable criticism and advice. We have now added an endoscopic image after HCD to Fig.2.

Reviewer 2.

  1. Figure 4, ultrastructural tight junction (TJ) analysis. At lines 359 and 372, please specify ‘intestinal’ villi. At first reading, focusing on the images shown in Figure 4, I understood ‘microvilli’ and did not figure out what the author meant in their description.

AUTHORS:

  1. We have now added “intestinal” villi to the sentences at line 359 and 372 to clarify the description. We have also added an arrow to the image pointing to the TJs in Fig.4.

Reviewer 2.

  1. Line  397: unclear sentence.

AUTHORS:

  1. Thanks for your attention. I agree that the sentence was unclear and have now changed it to “Mouse small intestine mounted in Ussing chamber treated with HG significantly increased Rep in preparations after HFD but not in preparations after NC”.

Reviewer 2.

  1. Line 442: osmium tetroxide?

AUTHORS:

  1. Quite right, it should be osmium tetroxide. We have now changed this at line 442.